

# Diabetes self-care and its associated factors among type 2 diabetes mellitus with chronic kidney disease patients in the East Coast of Peninsular Malaysia

Siti Aisyah Ramli, Nani Draman, Juliawati Muhammad and Siti Suhaila Mohd Yusoff

Department of Family Medicine, School of Medical Sciences, Health Campus, Universiti Sains Malaysia, Kubang Kerian, Kelantan, Malaysia

## ABSTRACT

**Introduction.** Diabetes self-care among diabetic patients is crucial as it determines how patients care for their illness in their daily routine for better diabetes control. This study aims to calculate the average score for diabetes self-care among patients with type 2 diabetes mellitus and chronic kidney disease and to identify factors that are associated with this score.

**Materials and Methods.** This cross-sectional study enrols patients over 18 years old with type 2 diabetes mellitus and chronic renal disease with an eGFR of less than 60 mL/min/1.73 m$^2$ in a tertiary hospital in Malaysia. The Malay version of the Summary of Diabetic Self-Care Activities (SDSCA) was used to assess diabetes self-care, the Malay version of the diabetes-related distress questionnaire (DDS-17) was used to assess diabetes distress, and the Malay version of the Patient Health Questionnaire-9 (PHQ-9) was used to assess depression. Data analysis was performed using both simple and multiple linear regression models to determine the associations between variables.

**Result.** One hundred and seventy-six eligible patients were recruited for this study. The mean score for diabetes self-care is 3.62. The eGFR ($p = 0.002$) and diabetes distress ($p = 0.004$) are the significant associated factors for diabetes self-care among type 2 diabetes mellitus patients with chronic kidney disease.

**Conclusion.** The mean score for diabetes self-care indicated a moderate level of self-care. The eGFR level and diabetes distress were important factors influencing diabetes self-care practices.

Corresponding author
Nani Draman, drnani@usm.my

## INTRODUCTION

The prevalence of diabetes mellitus (DM) worldwide is rapidly increasing, impacting over 380 million people (*Montero et al., 2016*). Chronic kidney disease (CKD) affects up to 8% to 16% of the global population, with diabetes mellitus, hypertension, and ageing being the leading causes (*Ang, 2022*). CKD is characterized by a low glomerular filtration rate or

albuminuria (*Chen et al., 2021*). In Malaysia, diabetes is a significant factor in the increased incidence of end-stage renal failure CKD.

According to the 2019 National Health and Morbidity Survey, the prevalence of diabetes increased in Malaysia from 13.4% in 2015 to 18.3% in 2019, with 3.6 million individuals (18 years of age and older) having the disease and 49% of cases going undiagnosed (*Institute for Public Health, 2020*). The prevalence of chronic kidney disease (CKD) has increased in Malaysia from 9.07% in 2011 to 15.48% in 2018 (*Institute for Public Health, 2020*). The rising trend in diabetes and its complications in Malaysia highlights the need for targeted screening and timely intervention to reduce the burden of the disease (*MOH, 2018*). Diabetic kidney disease (DKD) is due to excess glucose in the body, which can cause arterial disease and cardiovascular and renal disease (*Bikbov et al., 2020*; *Fraser & Roderick, 2019*). End-stage kidney disease (ESKD) is a progressive illness that can cause permanent renal impairment and early death in people with type 2 diabetes mellitus. However, its progression can be delayed with proper diabetes treatment and management (*Hussein et al., 2016*). Thus, diabetes self-care may help prevent complications and slow the progression of the disease to ESKD (*MacIsaac, Jerums & Ekinci, 2017*).

Diabetes self-care refers to daily routine activities to control diabetes. It is a process of understanding oneself and the role that diabetes plays in their lives. The goal of diabetes self-care is to alter behaviour for improved disease management and results. It is a lifelong commitment that requires knowledge, awareness, and self-empowerment. Clinicians may find the suggested diabetic self-care techniques helpful in managing specific patients, for example, by incorporating personalized dietary plans, regular physical activity regimens, consistent blood glucose monitoring, medication adherence, stress management techniques, and regular follow-up appointments to track progress and make necessary adjustments. Numerous research studies and recommendations in the field of diabetes support these methods (*Chima et al., 2021*). Numerous diabetes self-care initiatives have been documented in the literature but with varying degrees of success (*Kartika, Widyatuti & Rekawati, 2021*). If not, the measures have been linked to the improvement of clinical results and might save the burden of managing complications of diabetes mellitus (*Zerihun Sahile, Benayew Shifraew & Zerihun Sahile, 2021*).

Various factors can influence self-care behaviours, including knowledge of diabetes, physical activity, social support, and self-efficacy (*Zerihun Sahile, Benayew Shifraew & Zerihun Sahile, 2021*). Moreover, the opinions of medical experts suggested that clinical and sociodemographic factors such as age, gender, socioeconomic status, comorbidities, access to healthcare services, health literacy, social support, cultural beliefs, and psychological well-being had an impact on self-care maintenance, monitoring, management, and confidence (*Ausili et al., 2018*). Furthermore, adherence to self-care techniques among patients with type 2 diabetes has been linked to variables including sex, age, living situation, subjective health status, and length of diabetes (*Xie et al., 2020*). It is essential to consider these various factors when developing strategies to support and improve diabetes self-care practices.

Diabetic distress can lead to anxiety, conflict, frustration, and discouragement, which can impact the patient's ability to solve problems required to practice self-care. This may result in less effective self-care routines, eventually affecting glycaemic control (*Abd

*El Kader et al., 2023*; *Bala et al., 2021*). Practical strategies to address diabetes distress include referring patients to diabetes self-management education and support programs tailored to individual needs (*Owens-Gary et al., 2018*). Peer support, health coaching, and interventions that enhance self-efficacy and resilience have also been identified as beneficial approaches to mitigate diabetes distress and improve self-management behaviours (*Chima et al., 2021*).

Understanding and supporting people with diabetes involves considering various factors. These include knowing about diabetes, staying active, having support from friends and family, managing costs, and dealing with changing habits (*Reshma et al., 2021*). Living with diabetes can also bring up emotional struggles, known as diabetic distress, like worrying about health risks or feeling anxious about managing the condition. Moreover, the need for complex treatments like insulin and having good support from loved ones, especially spouses, can affect how someone copes emotionally (*Fisher, Polonsky & Hessler, 2019*). Complications from diabetes can also add to these emotional challenges. By recognizing and addressing these factors, we can better help people with diabetes care for themselves and improve their well-being (*Chew, Mohd-Sidik & Shariff-Ghazali (2015)*).

Patients with CKD and ESKD are more likely to experience depression, which is linked to a lower quality of life and a higher death rate. This is because dialysis induces psychological and physiological alterations. Patients with advanced CKD face challenges in performing self-care tasks effectively due to decreased exercise tolerance, functional capacity, and muscular atrophy. Moreover, these patients may require more care than those without CKD, impacting self-care practices. Thus, this study aims to determine the mean diabetes self-care score and associated factors among type 2 diabetes mellitus patients with chronic kidney disease (*Goh & Griva, 2018*).

## MATERIALS & METHODS

This cross-sectional study enrolled patients over 18 years old with type 2 diabetes mellitus and chronic renal disease (eGFR <60 mL/min/1.73 m$^2$) from a tertiary hospital in Malaysia. Patients with type 1 diabetes mellitus, those who were pregnant, and those who had kidney anomalies, such as ectopia, hydronephrosis, irregular renal surface, small kidney, or renal duplication, were all excluded. The sample size to determine the associated factors that affect diabetic self-care practice among type 2 diabetes mellitus with chronic kidney disease patients is done by comparing the mean for Malay and non-Malay for categorical variables (*Devarajooh & Chinna, 2017*). The most significant sample size for this objective is for the variable ethnicity. The detectable difference was decided after considering its clinical importance. Taking the alpha of 0.05 and power of 80%, the associated factor ethnicity yielded the highest sample size. Therefore, the minimum sample size required was 160. However, after considering 10% dropout, the calculated sample size is 176 (*Devarajooh & Chinna, 2017*). This study was approved by the Human Research Ethics Commitee, Universiti Sains Malaysia on 1 August 2022 (JEPeM code: USM/JEPeM/22040273). It involves preparing a detailed proposal, submitting it for review by the board, making any necessary revisions, obtaining formal approval, and adhering to ethical standards throughout the study.

### Study tools

We used the Malay versions of the Summary of Diabetic Self-Care Activities (SDSCA), the Diabetes Distress Scale (DDS-17), and the Patient Health Questionnaire-9 (PHQ-9) to assess diabetes self-care, diabetes distress, and depression, respectively.

### Malay version of the Summary of Diabetes Self-Care Activities (SDSCA)

A self-report tool called the SDSCA is used to gauge how well a person has managed their diabetes over the past seven days. The 12-item original SDSCA was created to evaluate five components of the diabetes treatment plan: blood-glucose testing, exercise, medication, and general and targeted diets (*Toobert, Hampson & Glasgow, 2000*). In 2012, the Malay SDSCA version was developed for children and adolescents aged between 10 and 18 with type 1 and type 2 diabetes mellitus (*Jalaludin et al., 2012*). In 2015, the Malay version of SDSCA among adults with type 2 diabetes was developed with Cronbach's alpha between 0.651 and 0.905 (*Adam Bujang et al., 2016*). It consisted of 11 items under five domains: diet (four items), exercise (two items), sugar control (two items), foot care (two items), and smoking. The total score will be divided into 10 to get the mean score of how many days each respondent spent on self-care practices. The response options for each item range from 0 to 7, with higher scores denoting improved performance in self-care activities.

### Malay version of the diabetes-related distress score (DDS) questionnaire

The DDS consists of 17 items. The Malay version of the diabetes-related distress questionnaire was adopted from a local study performed among 262 type 2 diabetes mellitus patients in Klinik Kesihatan Salak, Selangor, Malaysia (*Chew et al., 2015*). There was a high internal consistency with Cronbach's alpha 0.940. The correlation coefficient between the items and the total varied between 0.537 and 0.791. The Likert scale, which runs from 1 (not a problem) to 6 (a very serious problem), is used by this instrument. The patient's score will be summed up and divided by the number of items on that scale. A degree of discomfort that warrants clinical care is defined as having a mean item score of $\geq 3$. A mean score of less than 2.0 overall, on the other hand, denotes minimal to no discomfort. The range of 2.0 to 2.9 denotes significant discomfort (*Chew et al., 2015*).

### Malay version of the Patient Health Questionnaire (PHQ-9)

The original PHQ-9 questionnaire was designed to help doctors detect depression and other mental illnesses that are often seen in primary care settings based on specific criteria. PHQ-9 is used to rank the severity of depressed symptoms and includes the nine criteria that are used to diagnose DSM-IV depressive disorders (*Kroenke, Spitzer & Williams, 2001*). The Malay version of PHQ-9 was validated among adults over 18 who attended a family medicine clinic at Hospital Universiti Sains Malaysia from August 2001 to July 2003 (*Azah, Ni & Shah, 2005*). With a Cronbach's alpha of 0.67, PHQ-9 has strong internal reliability. At an alpha of 0.73, the test-retest reliability was likewise good. It includes the whole PHQ nine-item depression module. It is using a Likert-type format for responses. The respondent selects the option most accurately describes their feelings during the previous

two weeks. Every question is rated on a range of 0 to 3, which indicates how severe the symptoms of depression are. The PHQ-9 score has a range of 0 to 27. The patient's level of depression increases with the score.

## Data collection procedure

The data collection procedure is based on the attendance list. The average number of CKD with type 2 diabetes mellitus patients who come for follow-up at the clinic is about 5 per day. The list of the patients who will come to the next day's follow-up will be determined. The patient who will not turn up during the visit will be called up. On the day before each appointment, an attendance list is prepared. We review each patient's medical records from the clinic to obtain health information, including HbA1c levels, body mass index (BMI), and estimated glomerular filtration rate (eGFR). From this list, eligible patients are systematically and randomly selected in a 1:2 ratio for participation in the study. When patients arrive for their follow-up, they are informed about the study objectives and the confidentiality of the information they provide. After obtaining their consent, participants are given self-administered questionnaires, which they can complete in approximately 15 min. The research assistant will assist illiterate respondents in completing the questionnaires, and the "Respondent Information Sheet and Consent Form" will be explained to them and signed once they have understood the study's purpose and agree to participate.

## Data analysis

Data were entered and analyzed using SPSS Statistics version 26. Categorical data will be presented as frequency (n) and percentage(%). Based on their normality distribution, numerical data is presented as mean (SD).

Simple linear regression was used in an exploratory analysis to look for potential risk variables for practising effective diabetes self-care. To identify the associated determinants for effective diabetic self-care, multiple linear regression was performed on all variables with $p$ values less than 0.25 and clinically relevant variables. Concurrently, the model's other confounders were under control. The mean diabetic self-care score served as the dependent variable.

## RESULTS

This study comprised 176 eligible chronic kidney disease patients with type 2 diabetes mellitus, yielding a 100% response rate. The respondents' mean (SD) age was 61.6 (11.5) years. The respondents consisted of 51.1% of men and 48.9% of women. The majority were Malays, 170 (96.6%), married (85.2%) and employed (83.0%). The medical characteristics of participants are also shown in Table 1. The majority have a family history of diabetes mellitus (60.8%), 69.3% have complications of diabetes mellitus, and 87.5% have comorbidities. Mean HbA1c of the respondents was 8.53 (2.3). Table 1 provides an overview of the clinical and sociodemographic information.

The mean score for diabetic self-care in this study is 3.61 (1.39), which shows moderate diabetes self-care, which represents an average level of how well individuals are performing various self-care activities related to their condition.

**Table 1  Sociodemographic and clinical data of the participants ($n = 176$).**

| Variables | Mean (SD) | n (%) |
|---|---|---|
| Age (year) | 61.6 (11.5) | |
| Gender | | |
|     Male | | 90 (51.1) |
|     Female | | 86 (48.9) |
| Ethnicity | | |
|     Malay | | 170 (96.6) |
|     Chinese | | 4 (2.3) |
|     Others | | 2 (1.1) |
| Marital status | | |
|     Married | | 150 (85.2) |
|     Single/Divorced | | 26 (14.8) |
| Educational status | | |
|     None | | 11 (6.3) |
|     Primary | | 29 (16.5) |
|     Secondary | | 90 (51.1) |
|     Tertiary | | 46 (26.1) |
| Employment | | |
|     Unemployed | | 30 (17.0) |
|     Employed | | 146 (83.0) |
| Household income (RM) | RM 2400 (2900) | |
| Duration of diabetes mellitus (years) | 12.01 (8.9) | |
| Family history of diabetes mellitus | | |
|     Yes | | 107 (60.8) |
|     No | | 69 (39.2) |
| Complications of diabetes mellitus | | |
|     Yes | | 122 (69.3) |
|     No | | 54 (30.7) |
| Comorbidities | | |
|     Yes | | 154 (87.5) |
|     No | | 22 (12.5) |
| Treatment | | |
|     Oral medications | | 70 (39.8) |
|     Oral and insulin | | 59 (33.5) |
| Insulin | | 47 (26.7) |
| HbA1c (%) | 8.53 (2.3) | |
| eGFR (mL/min/1.73 $m^2$) | 36.7 (20.8) | |
| BMI (kg/$m^2$) | 25.9 (5.4) | |
| CKD stages | | |
|     3a | | 84 (47.7) |
|     3b | | 26 (14.8) |
|     4 | | 23 (13.1) |

**Table 1** (*continued*)

| Variables | Mean (SD) | n (%) |
|---|---|---|
| 5 | | 43 (24.4) |

**Notes.**

BMI, body mass index; eGFR, estimated glomerular filtration rate; HbA1c, glycated haemoglobin.

The significant association factors based on simple and multiple linear regression were shown in Table 2. Those display the relevant factors associated with diabetic self-care. eGFR and diabetes distress are significantly associated with diabetes self-care. The mean scores of diabetes self-care would rise by 0.016 for every unit increase in eGFR, and there was a statistically significant association between the two (95% CI = 0.006, 0.027, $p = 0.002$). If the diabetes distress score is increased by 1 point, the mean diabetes self-care score will decrease by 0.423 (95% CI = $-0.711$, $-0.135$, $p = 0.004$). The SDSCA mean score of 0.423 will be lower in those more distressed by diabetes. The model explains 10.6% of the variation in diabetes self-care mean score in the study sample ($R^2$) = 0.106, as shown in Table 2.

## DISCUSSION

This study focused on evaluating diabetes self-care practices, associated variables, and depression in patients with type 2 diabetes mellitus (T2DM) who also suffer from chronic kidney disease (CKD). Depression frequently exacerbates chronic kidney disease, a common consequence in people with type 2 diabetes mellitus and a significant barrier to efficient diabetic self-care management. Notably, no previous study conducted in Malaysia has explicitly addressed this patient population; hence, this research addresses a key need.

According to our study, the mean score for diabetic self-care was 3.61 (1.39), which was moderate. Our study mean was higher compared to a local study done in a rural area of Northern Peninsular Malaysia (*Yee, Salmiah & Rosliza, 2018*), while our research was conducted in the urban area of Kelantan. It was characterized by the higher level of knowledge held by urbanites relative to rural residents. Compared to urban regions, rural residents have reported a higher prevalence of self-care disabilities; nevertheless, this difference is gradually near (*Guo et al., 2022*). Access to diabetes self-management education also differs between urban and rural areas, with rural beneficiaries being less likely to participate in such programs home-based interventions have shown effectiveness, especially in elderly individuals with diabetic foot ulcers (*Kartika, Widyatuti & Rekawati, 2021*) and peer-support-based programs which have been developed to provide ongoing support through user-friendly technologies and trained peers, contributing to improved self-care behaviors and quality of life (*Luo et al., 2022*). The literature shows that rural individuals are at a higher risk of diabetes and tend to receive less access to medical facilities, have low health literacy, and have poor perceptions regarding health-related quality of life, leading to worse outcomes such as a higher prevalence of chronic complications (*Chen et al., 2023*).

However, the mean score of a local Selangor study involving patients with type 2 diabetes was 3.87, which was higher than our study (*Devarajooh & Chinna, 2017*). That study was done in the biggest district area in Selangor, which recruited patients from six primary

**Table 2  Associated factors for diabetes self-care among type 2 diabetes mellitus patient with chronic kidney disease using simple linear regression and multiple linear regression.**

| Variables | Diabetes self-care | Simple linear regression | | | Multiple linear regression | | |
|---|---|---|---|---|---|---|---|
| | Mean (SD) | $b^a$ (95% CI) | t stat | *p*-value | $b^b$ (95% CI) | t-stat | *p*-value |
| Age | | −0.013 (−0.031, 0.005) | −1.397 | 0.164 | −0.006 (−0.025, 0.012) | −0.692 | 0.490 |
| Gender | | | | | | | |
|   Male | 3.59 (1.43) | 0.039 (−0.378, 0.457) | 0.186 | 0.835 | | | |
|   Female | 3.63 (1.37) | | | | | | |
| Ethnicity | | | | | | | |
|   Malay | 3.59 (1.39) | 0.343 (−0.121, 0.807) | 1.459 | 0.146 | 0.177 (−0.272, 0.625) | 0.777 | 0.438 |
|   Non-Malay | 4.07 (1.78) | | | | | | |
| Educational level | | | | | | | |
|   None and primary | 3.28 (1.48) | 0.327 (0.078, 0.576) | 2.588 | 0.010 | 0.208 (−0.044, 0.459) | 1.631 | 0.105 |
|   Secondary andtertiary | 3.71 (1.37) | | | | | | |
| Marital status | | | | | | | |
|   Married | 3.64 (1.41) | −0.189 (−0.777, 0.398) | | 0.526 | | | |
|   Single/Divorced | 3.45 (1.33) | | | | | | |
| Employment | | | | | | | |
|   Unemployed | 3.61 (1.43) | −0.032 (−0.587, 0.523) | | 0.910 | | | |
|   Employed | 3.64 (1.24) | | | | | | |
| Household income(RM) | | 0.034 (−0.038, 0.106) | 0.943 | 0.347 | | | |
| Duration of diabetes mellitus | | −0.002 (−0.028, 0.024) | −0.155 | 0.877 | | | |
| | Mean (SD) | $b^a$ (95% CI) | t stat | *p*-value | $b^b$ (95% CI) | t-stat | *p*-value |
| Family history of diabetes mellitus | | | | | | | |
|   Yes | 3.75 (1.42) | −0.356 (−0.780, 0.068) | | 0.099 | −0.272 (−0.698, 0.153) | −1.263 | 0.208 |
|   No | 3.39 (1.35) | | | | | | |
| Complications of diabetes mellitus | | | | | | | |
|   Yes | 3.49 (1.43) | 0.396 (−0.053, 0.845) | 1.742 | 0.083 | 0.317 (−0.141, 0.774) | 1.366 | 0.174 |
|   No | 3.89 (1.30) | | | | | | |
| Comorbidities | | | | | | | |
|   Yes | 3.54 (1.39) | 0.602 (−0.022, 1.227) | 1.903 | 0.059 | 0.451 (−0.183, 1.085) | 1.405 | 0.162 |
|   No | 4.14 (1.37) | | | | | | |
| Treatment | | | | | | | |
|   Oral medications | 3.44 (1.53) | | | | | | |

**Table 2** (*continued*)

| Variables | Diabetes self-care Mean (SD) | Simple linear regression | | | Multiple linear regression | | |
|---|---|---|---|---|---|---|---|
| | | $b^a$ (95% CI) | t stat | *p*-value | $b^b$ (95% CI) | t-stat | *p*-value |
| Oral and insulin | 3.91 (1.17) | 0.046 (−0.213, 0.305) | 0.350 | 0.727 | | | |
| Insulin | 3.48 (1.42) | | | | | | |
| HbA1c(%) | | 0.052 (−0.038, 0.141) | 1.139 | 0.256 | | | |
| eGFR(mL/min/1.73 m$^2$) | | 0.016 (0.006, 0.027) | 3.197 | 0.002 | 0.016 (0.006, 0.027) | 3.197 | 0.002 |
| BMI(kg/m$^2$) | | −0.006 (−0.450, 0.330) | −0.313 | 0.755 | | | |
| Diabetes distress | | −0.466 (−0.759, −0.174) | −3.145 | 0.002 | −0.423 (−0.711, −0.135) | −2.903 | 0.004 |

**Notes.**

BMI, body mass index; eGFR, estimated glomerular filtration rate; HbA1c, glycated haemoglobin.

Forward multiple linear regression method applied. Model assumptions are fulfilled. There were no interactions among independent variables. No multicollinearity was detected (VIF less than 10). The assumption was checked, and there was no violation. Coefficients of determination ($R^2$) = 0.106.

There is a significant linear relationship between eGFR and diabetes self-care mean score.

health clinics and used a bigger sample size than in our study. Most study participants from that local study were female, accounting for 62% of the total participants, and had fewer diabetic complications compared to our study. According to a study among the Chinese population in China, female patients have better control of diabetes than males and have significantly better self-care and exercise habits than males (*Tang et al., 2021*).

On the other hand, participants in our study had a mean age of 61 years old. They exhibited better diabetes self-care compared to participants in a local study conducted in Northern Peninsular Malaysia, where the majority fell within the age range of 40–60 years old, indicating a younger population than ours. It correlated with a previous study among the Chinese population in China, which showed that patients more than 60 years old account for significantly better diabetic self-care practice in the diet compared to patients younger than 60 years (*Tang et al., 2021*). Older adults often have accumulated knowledge and experience managing their condition, leading to improved self-care behaviours (*Weinger, Beverly & Smaldone, 2014*). Additionally, elderly diabetic patients with higher social support tend to engage in better self-care practices, highlighting the importance of a supportive environment in enhancing self-care (*Kaya & Caydam, 2019*). Furthermore, among elderly low-income patients, health literacy, diabetes knowledge, exercise, and experiences with diabetes education are linked to higher diabetes self-care activities (*Jeong, Park & Shin, 2014*).

In our study, eGFR was statistically significantly associated with diabetes self-care activities, indicating that higher eGFR (>mL/min/1.73 m$^2$) is associated with improved diabetes self-care. Conversely, those who have superior kidney function also take better care of their diabetes. This is aligned with a study examining the relationship between patient characteristics and self-care in CKD patients with type 2 diabetes mellitus (*Zimbudzi et al., 2017*). One literature showed that lower scores in one or more self-care components were linked to an advanced stage of CKD. The study found less exercise and foot checks were

linked to more severe CKD. Individuals with chronic kidney disease (CKD) are less able and fit physically, which is important for both kinds of activities. Individuals with advanced chronic kidney disease (CKD) who have an eGFR of less than 60 mL/min/1.73 m$^2$ are less functionally capable, have more significant muscular atrophy, and find it challenging to perform self-care tasks (*Roshanravan, Gamboa & Wilund, 2017*).

Our study demonstrated that diabetes distress and diabetes self-care are significantly associated with glycemic control and overall quality of life in individuals with diabetes. Specifically, higher levels of diabetes distress were negatively correlated with better glycemic control, while improved diabetes self-care practices were positively associated with better glycemic outcomes and enhanced quality of life. Those who are more distressed will take less care of their diabetes. Diabetes distress is the emotional side of having the condition, characterized by anxieties, concerns, fears, and perceived threats of managing a chronic illness like diabetes over time (*Fisher, Polonsky & Hessler, 2019*). Diabetes distress has been persistently associated with suboptimal self-care practices in those living with the disease (*Chima et al., 2021*). It has been shown that poor self-care is linked to depression and diabetes distress, with diabetes distress being particularly associated with elevated HbA1c levels (*Nanayakkara et al., 2018*). Additionally, it has been found that poorer quality of life, inadequate glucose management, and fewer self-care practices are all linked to diabetic distress (*Beverly et al., 2019*). Lower distress was substantially correlated with higher support satisfaction, indicating that social support may be able to reduce the negative impacts of diabetic distress on self-care (*Baek, Tanenbaum & Gonzalez, 2014*). Diabetes distress, despair, and fatalism are interrelated characteristics that impact glycaemic control and self-care behaviours in people with type 2 diabetes (*Asuzu et al., 2017*).

Upon reviewing recent literature on diabetes self-care in T2DM patients with CKD, our study's findings remain valid. Recent studies have confirmed similar trends in self-care behaviours and their impact on disease management, reinforcing the reliability of our results. However, new research highlights additional factors, such as diabetes distress and eGFR, which could further influence self-care practices. Our study contributes to the current literature by comprehensively analysing self-care practices among T2DM patients with CKD. Our findings highlight those who have superior kidney function also take better care of their diabetes, and those who are more distressed will take less care of their diabetes. These insights are crucial for refining self-care guidelines and improving patient outcomes, especially in type 2 diabetes mellitus with chronic kidney disease.

The data used in this analysis only included type 2 diabetes patients attending Hospital Universiti Sains Malaysia in Kota Bharu for outpatient follow-up; other medical facilities in different Kelantan districts or other states in Malaysia were not included. Therefore, the study conclusions do not accurately reflect the whole population. In addition, the vast majority of participants were Malay. Consequently, the study conclusions do not adequately represent the diverse ethnic population in Malaysia. The results could be skewed by inherent biases in the selection technique, such as selecting individuals from specific healthcare institutions or those who practise self-care more proactively. The study may not have addressed every relevant component impacting diabetic self-care, which could result in an insufficient understanding of the problem.

This study provides valuable insights into the relationship between diabetes distress, self-care, and glycemic control within a specific region of Malaysia. However, its limitations include a small sample size and lack of ethnic diversity, which restricts the generalizability of the findings. Additionally, the study's focus on a limited set of variables and its cross-sectional design constrain its ability to capture the broader context and causal relationships over time. Despite these limitations, the study's strengths lie in its detailed examination of the impact of diabetes self-care on health outcomes and its relevance to local healthcare practices. Future research should address these limitations by incorporating larger, multiethnic samples, exploring a more comprehensive range of influencing factors, and employing longitudinal designs to provide a more comprehensive understanding of diabetes management.

## CONCLUSION

The average diabetes self-care score indicated a moderate level of self-care. The eGFR level and diabetes distress were significant factors influencing diabetes self-care practices. Patients experiencing greater diabetes distress tend to engage less in self-care activities. Clinicians should educate patients on stress management to improve their self-care practices.

## ACKNOWLEDGEMENTS

The authors would like to thank participants for their cooperation and willingness to join this study.

### Funding
This study was supported by a grant from Infinity Medical Sdn. Bhd. The funders had no role in study design, data collection and analysis, decision to publish, or preparation of the manuscript.

### Grant Disclosures
The following grant information was disclosed by the authors:
Infinity Medical Sdn. Bhd.

### Competing Interests
The authors declare there are no competing interests.

### Author Contributions
- Siti Aisyah Ramli conceived and designed the experiments, performed the experiments, analyzed the data, authored or reviewed drafts of the article, and approved the final draft.
- Nani Draman conceived and designed the experiments, performed the experiments, authored or reviewed drafts of the article, and approved the final draft.

- Juliawati Muhammad analyzed the data, prepared figures and/or tables, authored or reviewed drafts of the article, and approved the final draft.
- Siti Suhaila Mohd Yusoff conceived and designed the experiments, authored or reviewed drafts of the article, and approved the final draft.

## Human Ethics

The following information was supplied relating to ethical approvals (*i.e.*, approving body and any reference numbers):

The Human Research Ethics Commitee, Universiti Sains Malaysia granted Ethical approval to carry out the study within its facilities (Ethical Application Ref: USM/JEPeM/22040273)

## Data Availability

The raw data is available in the Supplementary File.

## Supplemental Information

Supplemental information for this article can be found online at http://dx.doi.org/10.7717/peerj.18303#supplemental-information.

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
