# Peer review of "Diabetes self-care and its associated factors among type 2 diabetes mellitus with chronic kidney disease patients in the East Coast of Peninsular Malaysia"

_PeerJ, doi:10.7717/peerj.18303_

## Round 0.1 · original submission · Minor Revisions

Please review the comments from the reviewers, and make edits to the document as the reviewers have recommended and then send us within 10 days for consideration of publication.

·

Basic reporting

Basic Reporting
• Language and clarity: The manuscript is written in clear, professional English. However, minor grammatical errors are present, which can be improved for better clarity and are indicated in specific comments.
• The research question is well-defined, relevant, and meaningful. It directly addresses the need to identify factors influencing diabetes self-care activities, which is crucial for improving the CKD patient outcomes.
• The plagiarism rate is very low which is much acceptable.
Abstract
• The abstract briefly summarizes the research, methods, results, and conclusions. Each section flows logically, helping in the reader's comprehension.
Introduction:
• The introduction provides a thorough context for the study, explaining the importance of diabetes self-care and the prevalence of diabetes and CKD in Malaysia. The background information is well-referenced and relevant to the study.
Materials and Methods
• The methodology is rigorously designed and described with sufficient detail to allow replication. The use of validated tools like the Malay versions of the SDSCA, DDS-17, and PHQ-9 enhances the reliability of the data collected. However why the author prefer linear regression for a given variables is not clear.
• The inclusion and exclusion criteria are clearly stated. However, it is important to clarify how participants were recruited and any potential biases this might introduce.
• Ethical approval and consent procedures should be clearly stated to ensure compliance with ethical standards.
• It is advisable to provide more detailed descriptions of the statistical methods used, including any assumptions made and how they were tested.
• The tables are well-labeled. However, comments have been provided under the "specific comment" section related to the content of Tables 3 and 4. These comments need to be addressed to improve the clarity, accuracy, and completeness of the data presented.
Conclusions:
• The conclusions are well-stated and linked to the original research question. They are appropriately limited to the supporting results.
• It would be beneficial to discuss the practical implications of the findings in more detail

Specific Comments
Title
• What is the rationale for selecting one independent variable (IV) for the title over others? The current title is descriptive but could be more concise. It would be better to align the title with your objective: “Diabetic Self-Care and Associated Factors in Type 2 Diabetes Patients with Chronic Kidney Disease in East Coast Peninsular Malaysia.” There is no need to select one independent variable over others for the title in this specific case, especially since you were unable to find an association with depression and are not testing depression against diabetic self-care. However, if you insist on including depression in the title, consider revising it to “Diabetic Self-Care and Depression in Type 2 Diabetes Patients with Chronic Kidney Disease in East Coast Peninsular Malaysia” for brevity and clarity.
Abstract
• Ensure consistency in subtitles between the abstract and the main body of the paper. For example, if the abstract uses “Background” and “Methods,” the main body should use the same terms instead of “Introduction” and “Materials and Methods.”
Objectives
• Lines 6-8: The sentence, "This study aims to…," is clear but could be more precise. Consider revising it to, “This study aims to determine the mean diabetes self-care activities score and its associated factors among type 2 diabetes mellitus (T2DM) patients with chronic kidney disease.”
Methodology
• Lines 130-132: The sentence, "This is a cross-sectional study, enrolling…." is wordy. Consider breaking it into two sentences or consider rewriting it again for clarity:
Suggestion: “This cross-sectional study enrolled patients over 18 years old with type 2 diabetes mellitus and chronic renal disease (eGFR < 60 mL/min/1.73 m2) from a tertiary hospital in Malaysia.”
• Lines 141-145: The sentence, "The Malay version of…….." is long and complex. Simplify it to: “We used the Malay versions of the Summary of Diabetic Self-Care Activities (SDSCA), the Diabetes Distress Scale (DDS-17), and the Patient Health Questionnaire-9 (PHQ-9) to assess diabetes self-care, diabetes distress, and depression, respectively.”
• Line 149: A self-report tool called the SDSCA is used to ………….. and Line 195-196: “After obtaining their consent, participants are given self-administered questionnaires, which they can complete in approximately 15 minutes.” Clarify how you handled respondents with no formal education, accounting for 6.3% of the sample, given that participants were required to complete self-administered questionnaires.
• Lines 192-193: Provide detailed information on participant recruitment to assess potential sample biases.
• Ethical Approval and Consent: Include detailed ethical considerations, specifying the IRB number or supplementary documentation. Explain how consent was obtained from participants without formal education.
Data Analysis:
• Lines 16-17: The statement, “Both simple and multiple linear regression were used to analyze the data,” can be expanded for clarity:
Suggestion: “Data analysis was performed using both simple and multiple linear regression models to determine the associations between variables.”
• Why did you prefer linear regression? Logistic regression might be more appropriate for your data. Even if linear regression is viable, your data is more suitable for a logistic regression model. Consider treating the DV (self-care) and all other IVs including depression as categorical variables and re-analyze using logistic regression, given the clinical classification of PHQ-9 scores is categorical and most of your IVs are categorical in the manuscript. You can also treat variables like HbA1c, eGFR, BMI as categorical.
• Line 135-137: Detail how you calculated your sample size.
• Lines 213-216: For objective three, the score on the Patient Health Questionnaire (PHQ) was the dependent variable. In my opinion, this is not appropriate for this particular manuscript and its scope. There is no indication of a third objective, and it is unnecessary. I recommend treating depression as one of your independent variables (IVs) and recalculating your simple and multiple linear regressions. See the details in the 'Results' section."
Results
• Your result interpretation generally needs more depth. I do not find your writing to be informative. Also your multiple linear regression analysis didn’t demonstrates a rigorous approach to exploring the relationships between variables.
• Line 213-237 …………………….as shown in Table 3.
So, in the case of Table 3, why did you only attempt to assess the association of only 2 IVs with that of DV? What are your selection criteria for candidates? You mentioned in the data analysis section of the method that 'multiple linear regression was performed on all variables with p-values less than 0.25 and clinically relevant variables.' You have many other variables that have a p-value of < 0.25?"
• Line 240-243: Table 4 demonstrated that according to a simple linear regression model, increasing diabetes self-care by 1 point would lower depression by 0.153; nevertheless, there was no statistically significant correlation between diabetic self-care and depression (p = 0.583, 95% CI = −0.704, 0.397).
Why do you need to examine the effect of self-care on depression, which is beyond the scope of your provided objectives in this manuscript? By doing so, you imply that you have two separate dependent variables. Instead, you could consider depression as the independent variable and examine its effect on self-care. so don’t initiate another inquiry by treating your first dependent variable as an independent variable and depression as the dependent variable."

Discussion
• Line 305: Rewrite the sentence
Suggestion: “Our study also indicated that diabetes distress and diabetes self-care were important.”
• Lines 348-350: "In comparison, patients ……..This sentence structure is uncomfortable.
Suggestion: "Patients experiencing greater diabetes distress tend to engage less in self-care activities. Clinicians should educate patients on stress management to improve their self-care practices."
Conclusion: The conclusion needs rewriting a gain
Suggestion: The average diabetes self-care score indicated a moderate level of self-care. The eGFR level and diabetes distress were identified as significant factors influencing diabetes self-care practices.

References
• Line 89: Did you use automatic reference styling, such as Mendeley software, or another tool? If so, why does reference number 14 appear before reference number 10 in the manuscript?

General comments
Strengths:
• The study addresses an important topic concerning diabetes self-care and associated factors among patients with chronic kidney disease, contributing to the understanding of this population's health management.
• The manuscript is well-organized, and the use of validated assessment tools adds credibility to the findings.
Weaknesses:
• The title could be more concise and aligned with the study objectives to improve clarity and relevance.
• The multiple linear regression analysis didn’t demonstrates a rigorous approach to exploring the relationships between variables.
• The interpretation of results lacks depth
Suggestions for Improvement:
• Refine the title
• Reanalyze the linear regression analysis
• Enhance the interpretation of results
• Ethical considerations should be more clearly stated.

Experimental design

No comment

Validity of the findings

No comment

Reviewer 2 ·

Basic reporting

This manuscript has been written well, and there are no significant issues with English or grammar.
The conclusion (in the abstract) can be reframed and contextualized.
All abbreviations, such as DKD and ESKD, must be spelled out when used for the first time on a page.
The study objectives can be made explicit with clear hypotheses presented at the beginning. Subsequently, a reference to statistically and non-statistically significant relationships can be made later.
The manuscript needs to be carefully evaluated for missing citations. For example – “Numerous research studies and recommendations in the field of diabetes support these methods. Numerous diabetes self-care initiatives have been documented in the literature but with varying degrees of success.” Reference to those studies must be provided.
I have included some other suggestions which can improve the manuscript.

Experimental design

The study could be informative and valuable if it includes the literature gap, indicating its rationale. However, the study has included sufficient literature to demonstrate the relationships between different variables to support the logical implications for the study’s overall goal. The overall goal should be rewritten. It says, “This study aims to determine the mean diabetes self-care score and its associated factors among type 2 diabetes mellitus patients with chronic kidney disease.” What does “the mean diabetes self-care score” indicate? It can be reframed. Also, what are those associated factors?

The result section should include more than just information about the tables. It should be a narrative description of the results with reference to hypotheses and appropriate tables for further information. Authors should make reference to their research questions/hypotheses.
Information in all tables must be interpreted, and key information should be presented in a descriptive manner for readers, such as what are the key characteristics of the research subjects and what the mean score of self-care actually represents. It should be more than just “moderate self-care.”
Authors claimed in line 310 “ There was no previous study done to determine a direct association between diabetes self-care and depression.” Please check the following study found through Google Scholar and they have used the same self-care scale. Authors need to do a literature review to contextualize the result. There should also be a rationale for difference or convergence with previous findings.

Tohid H, Papo M, Ahmad S, Sumeh AS, Jamil TR, Hamzah Z. Self-care activities among patients with Type 2 Diabetes Mellitus in Penampang, Sabah and its association with depression, anxiety and stress. Malaysian Journal of Public Health Medicine. 2019 Jan 1;19(1):117-25.
Additionally, in the same paragraph, there could be some citation issues.

Validity of the findings

The validity of the result must be reevaluated in light of a new literature review. How the findings contribute back to the literature must be presented succinctly.

---

## Round 0.2 · Minor Revisions

Please correct the following details:
* Please describe the Ethics Approval Process and state who provided Ethical approval for the study and what processes were followed to obtain that approval since you had participants who shared individual-level data. Please include the details of the Ethical Approval Authority and relevant numbers. If Ethics Approval was not obtained or were waived, please include that information and explain why that was the case.
* Please address the concerns of the first reviewer but you do not need to address why you used linear regression as your "dependent" variable was measured on a continuous scale, hence it explains.
* You have written, "The sample size to determine the associated factors that affect diabetic self-care practice among type 2 diabetes mellitus with chronic kidney disease patients is done by using comparing two means for categorical variables", you need to specify which two means?
* You have written in the methods section, "For objective three, the confirmatory analysis involves comparing the overall mean score of diabetes self-care practice among individuals with type 2 diabetes mellitus and chronic renal disease to depression using simple linear regression analysis. The score on the Patient Health
Questionnaire (PHQ) was the dependent variable", you have made no mention of Objective 3 anywhere in the paper nor have you presented any results of confirmatory factor analysis, please remove this paragraph, this was missed by the reviewers for some reason
* Please correct the grammatical errors as the manuscript must be written in past tense, not a mix of future and past tenses
* Please tidy up the tables and then resubmit.

·

Basic reporting

Review Report 2
While the manuscript is generally well-written, some minor grammatical, spelling, and sentence errors still need to be addressed. Additional language refinement is necessary to enhance overall clarity.
Examples
• "Our study also indicated that diabetes distress and diabetes self-care were important."
Important in what way? The sentence is vague and should specify what is meant by "important."
• "To the best of our abilities, this study was designed to obtain authentic data and trustworthy results."
Unnecessary: This sentence does not add meaningful content to the manuscript and could be removed.
• "These and other limitations are listed below."
Unnecessary: This phrase does not contribute valuable information and should be omitted for conciseness.
• In the discussion section, the limitations and strengths should be more clearly articulated to highlight the scope of the study's findings.
• Misplaced commas and spelling errors like 'melltus' should be corrected to “mellitus”

Experimental design

No comment

Validity of the findings

No comment

Additional comments

No comment

Reviewer 2 ·

Basic reporting

I am satisfied with the recent changes made by the authors.

Experimental design

I am satisfied with the recent changes made by the authors.

Validity of the findings

I am satisfied with the recent changes made by the authors.

---

## Round 0.3 · accepted · Accept

Thank you for your submission. I can confirm that as authors, you have addressed all of the reviewers' comments, I have assessed the revision myself and I am happy with the current version, and the manuscript is ready for publication.